# The Prevalence of Urinary Incontinence among Adolescent Female Athletes: A Systematic Review

**DOI:** 10.3390/jfmk6010012

**Published:** 2021-01-28

**Authors:** Tamara Rial Rebullido, Cinta Gómez-Tomás, Avery D. Faigenbaum, Iván Chulvi-Medrano

**Affiliations:** 1Tamara Rial Exercise & Women’s Health, Newtown, PA 18940, USA; rialtamara@gmail.com; 2Research Group Physiotherapy and Readaptation in Sport, Department of Physiotherapy, Catholic University of Murcia (UCAM), 3010 Murcia, Spain; 3Department of Health and Exercise Science, The College of New Jersey, Ewing, NJ 08628, USA; faigenba@tcnj.edu; 4UIRFIDE (Sport Performance and Physical Fitness Research Group), Department of Physical and Sports Education, Faculty of Physical Activity and Sports Sciences, University of Valencia, 46010 Valencia, Spain; ivan.chulvi@uv.es

**Keywords:** pelvic floor dysfunction, women’s health, pelvic floor training, youth

## Abstract

This review aimed to synthesize the most up-to-date evidence regarding the prevalence of urinary incontinence (UI) among adolescent female athletes. We conducted a systematic review of studies regarding UI in female athletes less than 19 years of age. This review was conducted in accordance with the Preferred Reporting Items for Systematic Reviews and Meta-analyses (PRIMSA). The electronic databases of PubMed, Embase, Cochrane Central Register of Controlled Trials (CENTRAL), Scopus, and Web of Science (WOS) were searched between October and November 2020. After blinded peer evaluation, a total of 215 studies were identified and nine were included. Risk of bias was assessed using the Strengthening the Reporting of Observational Studies in Epidemiology (STROBE) checklist. This review identified a prevalence of UI in adolescent female athletes between 18% to 80% with an average of 48.58%. The most prevalent sports were trampolining followed by rope skipping. The prevalence of UI among adolescent female athletes practicing impact sports was significantly prevalent. There is a need for further research, education, and targeted interventions for adolescent female athletes with UI.

## 1. Introduction

Urinary incontinence (UI) is defined as any complaint of involuntary loss of urine [1]. Mostly prevalent in women, the broad range of UI is 5–27% [2], with an average prevalence of 27.6% based on a review of population studies [3]. The most common type of UI is stress urinary incontinence (SUI) that is defined as any complaint of involuntary loss of urine on effort or physical exertion [1]. Strenuous exercise has been cited as a risk factor for developing symptoms of SUI [4]. Recently, a subcategory of athletic incontinence was proposed as a new term for a specific SUI that occurs during sport activities or competition [5]. One of the most prevalent pelvic floor dysfunctions reported in female athletes is SUI [6,7,8,9]. For instance, a meta-analysis that included 7507 women with age ranges between 12 and 69 years, found that the prevalence of SUI was 33.69% for the female athletes compared to 24.40% in the control group [10].

The younger female athletes seem to display isolated symptoms of pure stress UI which is an uncomplicated SUI without other symptoms of urge incontinence or bladder dysfunction [11]. High-impact sports involving jumping, landing or running have shown the highest prevalence rates of urinary loss among young female athletes [12,13,14,15]. A recent meta-analysis by Teixeira et al. found a 35% prevalence rate of UI in female athletes (average age of 23.8 years) practicing different sports. When compared with sedentary women, female athletes displayed a 177% higher risk of presenting with UI symptoms [16]. Moreover, female athletes practicing high-intensity activities displayed greater odd ratios of SUI symptoms than those practicing less intense physical activity [9,17]. Similar UI prevalence rates (25.9%) were described in a review with meta-analysis focusing on female athletes involved in high-impact sports such as volleyball, athletics, basketball, cross-country, skiing, and running [8]. UI during practice or competition can cause embarrassment and negatively impact athletic performance. It has been reported that a vast majority of female athletes (~80%) with UI are too embarrassed to tell their coaches, which sustains unawareness of the problem and delays intervention [18,19]. UI can affect an athlete’s quality of life and impact performance [20], leading to sport drop-out [15,21].

The underlying mechanisms by which young nulliparous female athletes show higher levels of UI as compared to their sedentary females [16,17] are still not scientifically understood. The continence mechanism during sports practice has been hypothesized to be affected by a variety of kinematic and sport-related factors such as pelvic floor displacement during jumps and running [22,23], neuromuscular fatigue of the pelvic floor muscles during strenuous physical activity [24], and morphological changes of the pelvic floor muscles [25]. Moreover, low energy availability, low body mass index (BMI), estrogen changes, and hypermobility joint syndrome have also been suggested as possible contributing factors for developing UI in female athletes [26,27].

Elite female athletes experiencing UI at an early stage are more likely to report UI symptoms later in life [7]. This is a condition that should be addressed early in life and studied in order to provide better care and support. To date, little is known about the pelvic floor function of young female athletes. Although previous systematic reviews have analyzed the incidence of UI in physically active and athletic females of all ages [4,8,10,16], no previous reports have focused their attention on adolescent female athletes. Given the unique developmental characteristics occurring during adolescence and the previously demonstrated association between high impact training and UI, the prevalence of UI in adolescent athletes needs to be specifically addressed. Our main goal was to identify the prevalence of UI in female athletes less than 19 years of age and provide an understanding of the types of sports associated with the highest prevalence rates.

## 2. Materials and Methods

### 2.1. Information Sources and Search

The conduct and reporting of this systematic review complied with the Preferred Reporting Items for Systematic review and Meta-Analyses (PRISMA) guidelines [28].

A systematic search of electronic databases including PubMed, Embase, Cochrane Central Register of Controlled Trials (CENTRAL), Scopus, and Web of Science (WOS) was carried out between October and November 2020 independently by two blinded authors. No restrictions on language or publication timeline were applied. The search strategy used keywords, mesh terms, and Boolean connectors (AND/OR) including: “Stress urinary incontinence” OR “urine loss” OR “pelvic floor muscles” AND sport OR athlete OR “female athlete”. Search results were limited to species (human) and age (birth–18 years) and source type (journals).

### 2.2. Eligibility Criteria and Study Selection

Retrieved titles and abstracts were assessed for eligibility for inclusion, and duplicate entries were removed. The same two authors independently reviewed the text of the studies for eligibility. Articles published up to November 2020 were eligible for inclusion. The criteria for inclusion were: (1) study participants included adolescent females participating in sport or athletic activities; (2) study provides an assessment of UI symptoms; (3) study published in a peer-reviewed journal in any language. Randomized controlled trials (RCTs) with two or more parallel groups and crossover trials, non-RCTs were eligible for inclusion if they met the previously mentioned criteria. The criteria for study exclusion were: (1) participants > 19 years old; (2) participants who underwent any type of pelvic floor surgery; (3) participants during their pregnancy and postpartum period and; (4) systematic review, meta-analysis, or case study.

### 2.3. Data Collection Process and Quality Assessment

For each study, data were extracted on the characteristics of the population and intervention such as: (1) last name of the first author; (2) years of publication; (3) study design; (4) sample characteristics (age, sample analyzed, weight, body mass index, sport practice, and hours of weekly training); and (5) instrument assessing symptoms of UI. Risk of bias was assessed independently by two authors using the Strengthening the Reporting of Observational Studies in Epidemiology (STROBE) checklist [29]. The same two researchers rated the studies and discrepancies were resolved by consensus. Data reporting completeness was assessed by applying the STROBE cross-sectional checklist reporting classified as “not reported or unclear”, “some information mentioned but insufficient”, or “clear and detailed information provided”.

## 3. Results

### 3.1. Study Selection

The search strategy yielded 500 potentially relevant studies. After the removal of duplicates, 321 records were screened. Of those, 215 potential titles were selected after the database filter insertion. Among those, only nine studies met the criteria for inclusion and were selected for analysis in this systematic review. The study selection flow chart is shown in Figure 1.

### 3.2. Overview of Study Characteristics

Table 1 provides an overview of the characteristics of the studies included in this review. Table 2 provides the participants’ characteristics of weight, body mass index (BMI), and hours of training per week.

Our systematic review identified nine studies published between 2002 and 2020. The total sample was 633 female athletes, with an average age of 16.15 years, BMI ranging from 18.9 to 21.7 kg/m^2^, and 6–19 h of training per week. We calculated a mean of prevalence of 48.58% for all the samples that were involved in different sports. Almost all study designs were cross-sectional (*n* = 8) where one had a pilot cross-sectional design. The risk of bias was assessed with the STROBE checklist for cross-sectional studies [29]. Figure 2 presents a heat map showing the grading of reporting completeness and quality for selected items according to the Strengthening the Reporting of Observational Studies in Cross-sectional studies. Eighty-seven percent of the articles explained the scientific background and rationale for the investigation and 62% stated specific objectives, including any specified hypotheses. Only 50% of the studies presented key elements of study design early in the paper and described the setting, locations, and relevant dates, including periods of recruitment, exposure, follow-up, and data collection. Study size was only explained in one study [31]. Clarity in defining all outcomes, exposures, predictors, potential confounders, and effect modifiers was applicable for 75% of the studies. Fifty percent of the included studies explained all of the statistical methods, including those used to control for confounding variables. Lastly, all studies summarized key results with reference to study objectives and discussed limitations of the study, taking into account sources of potential bias or imprecision.

### 3.3. Principle Findings

This systematic review identified a range of UI prevalence rates ranging from 18.2% to 80% and yielding a mean prevalence of 48.58%. In reports that assessed UI in one specific sport, the highest prevalence rates were found in trampolining (80%) followed by rope skipping (75%) and soccer (62.8%). On the other hand, the lowest rates of UI were found in practitioners of rhythmic gymnastics (31.8%).

The main outcome for assessing UI symptoms was the International Consultation on Incontinence short form questionnaire (ICIQ-SF), which was used in 5 of the 9 studies. Only one study used a quantitative measurement of UI through the pad-test [31]. Almost all studies included secondary assessments with questionnaires regarding the impact of UI on quality of life, specific type of urine loss, or associated pelvic floor dysfunctions such as fecal incontinence, sexual dysfunction, and pelvic organ prolapse. Only one study [31] measured muscular strength of the pelvic floor muscles. Of note, two of the included studies assessed female athlete triad risk factors including disordered eating behaviors [19,20]. Two studies assessed athletes’ knowledge about pelvic floor muscle training (PFMT) [18,20]. A high percentage of adolescent female athletes (69% to 90%) had never heard of PFMT [18,20]. Moreover, 87% of adolescent female athletes stated they would not mention their UI symptoms to their coach [19].

## 4. Discussion

The aim of the present review was to systematically review the prevalence of UI among adolescent female athletes. Notably, we found a wide range of UI prevalence rates among young female athletes varying from 18% to 80%, with an average prevalence of UI symptoms in female adolescent athletes about 50%. Our results are slightly higher than the meta-analytic data presented by Teixeira et al. [16] for female athletes with an average age of 23.8 years, with a weighted average of 36% of UI prevalence. Additionally, our findings are significantly higher than the study by Hagovska et al. [34] who reported a UI prevalence of 14.3% in 503 adult female athletes (21.1 ± 3.6 years of age) who participated in high-impact sports. Notably, in the aforementioned study, the authors determined the impact of each sport activity based on metabolic intensity rather than on ground impact forces [17,35]. Along these lines, our data are in the range reported by Bø who reported a UI prevalence range between 10% and 55% in female athletes between 15 and 64 years of age [15]. Another review involving female athletes between 12 and 45 years [10], noted average prevalence rates varied from 1% to 42.2%

Our review included a total sample of 633 young nulliparous female athletes practicing a wide range of sports. Several studies included samples of athletes practicing different sports. We applied a classification of sport impact based on the study by Groothaussen and Siener [30] that has been specifically applied to the analysis of the impact of sports on the pelvic floor [7,10]. This impact classification is divided in 4 distinct groups: impact grade 3 (>4 times body weight, e.g., jumping); impact grade 2 (2–4 times body weight, e.g., sports involving sprinting activities and rotational movements), impact grade 1 (1–2 times body weight, e.g., such as lifting light weights); and impact grade 0 (<1 time body weight, e.g., swimming). The highest rates of UI in our sample were of grade 3 sports, which included jumping and landing actions (i.e., trampolining and rope skipping). Team sports graded 2 such as soccer, basketball, and track and field were found to display high prevalence rates as well. Impact activities such as running, jumping, and landing have been associated with increased intra-abdominal pressure in the pelvic organs and tissues [22,23]. The additional ground reaction forces placed on the continence structures may lead to displacement or insufficient counteractive muscle activity of the pelvic floor [22]. Another possible mechanism that may explain these prevalence rates is the relatively high metabolic intensity of selected sporting activities that contributes to the possible neuromuscular fatigue displayed by the pelvic floor muscles during training or competition [24]. Overall, the main characteristic of all sports performed in our sample was an impact grade between 2 and 3 [30].

The benefits of sports practice early in life are well established; however, young female athletes are not immune to suffering sport-related injuries or illness [36]. Particularly, the young female athlete can suffer from pelvic floor dysfunctions such as UI as well as pelvic pain and anal incontinence [6,34]. Almeida et al. [34] reported fecal incontinence, dyspareunia, and difficulty emptying the bladder in the female athletic group [34]. Low energy availability in female athletes has been noted as another health impairment that can impair pelvic floor function due to a constellation of hormonal, metabolic, and neuromuscular imbalances [26]. In this sense, Whitney et al. [37] found that female adolescent athletes (aged 15 to 19 years) with low energy availability had a higher prevalence of UI when compared with those with adequate levels of energy. Two studies included in our review assessed for the presence of eating disorders [19,20]. Parmigiano reported that 15% of their sample was at risk for suffering an eating disorder and Gram and Bø noted that 9.3% of adolescent rhythmic gymnasts were at risk for disordered eating [20]. In our review, the average volume of training and BMI of the sample ranged from 18.9 to 21.7 kg/m^2^ and 6 to 19 h of training per week. Collectively, these observations suggest that the high volume and intensity of training along with low energy availability could be potential risk factors for developing UI in adolescent female athletes.

Bø and Sundgot-Borgen described that the presence of UI early in life is a strong predictor for UI later in life (ORR of 8.57) [7]. Moreover, leakage during sport practice has been shown to be a barrier to sports participation for young females [15,21]. Due to the observable health and fitness benefits of sports participation for girls and young women [36,38], additional studies are needed to improve our knowledge regarding pelvic floor dysfunction and implement effective preventative measures in active females. There is a lack of data targeting adolescent females investigating preventative, educational, and treatment modalities for UI. Given the high prevalence of UI in young female athletes and the lack of awareness of evidence-based preventative neuromuscular strategies such as PFMT and pelvic floor therapy [18,20,27], more studies are warranted. Pelvic physiotherapy has been found to be more effective in achieving continence in elite female athletes and pregnant athletes engaged in aerobic exercise compared to non-athletes [27]. For all these reasons, we suggest early screening with specific evaluation tools such as the pre-participation gynecological evaluation of female athletes proposed by Parmiagiano et al. [19] as well as the incorporation of specific neuromuscular training programs for the pelvic floor [13]. Increased awareness and educational programs targeting coaches and all female athletes regarding the pelvic floor musculature and specific dysfunctions such as UI are also warranted.

Limitations of this review are the small sample size, heterogeneity, and variability of outcome measures as well as the lack of reliable quantitative outcome measures for UI. The selected studies used validated questionnaires to assess urinary symptoms in young athletes. However, these questionnaires were validated in adult populations. More reliable diagnostic outcomes would improve the quality of the studies. In addition, the analysis of co-founding factors specific to the female adolescent athlete such as menstrual cycle and nutritional status would improve the quality of the studies. We recommend the use of the STROBE checklist for risk of bias study assessment to improve the scientific report of these studies and a classification of sport characteristics and impact, which would additionally improve their comparison and assessment. The development and validation of a specific questionnaire for assessing UI symptoms in adolescent females is warranted.

## 5. Conclusions

UI during exercise and sports is a concern for young female athletes. Our findings highlight a 48.8% prevalence rate among adolescent female athletes where practitioners of high-impact sports show the highest prevalence rates. Given the high prevalence of UI among adolescent female athletes involving impact sports graded 2 and 3, concerted efforts are needed to provide early education and implement prevention measures before young female athletes experience the burden of UI. Future research is needed to guide our understanding of the underlying physiopathology and unique characteristics of the adolescent female athlete’s pelvic floor muscle activity during impact sports.

## Figures and Tables

**Figure 1 jfmk-06-00012-f001:**
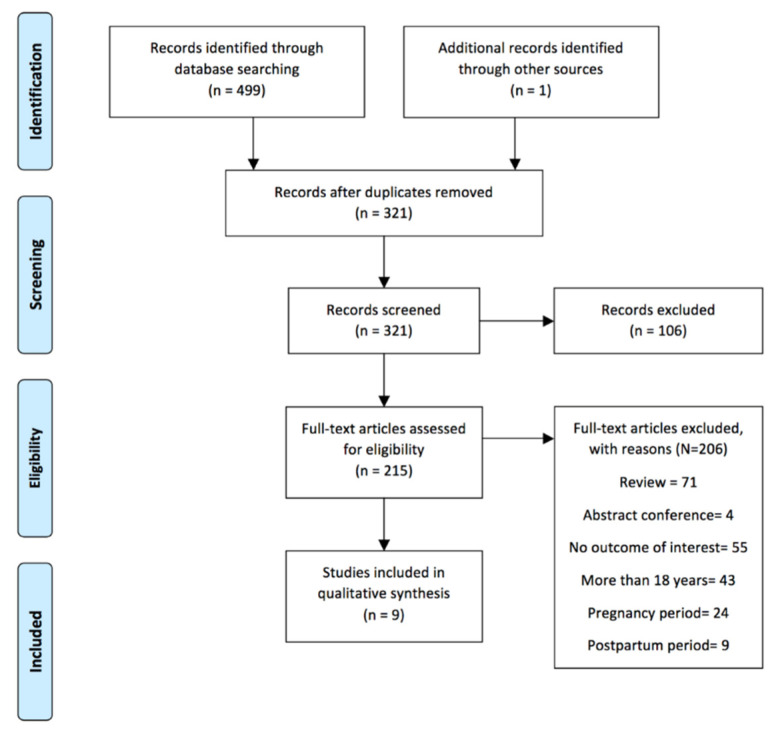
Flow diagram for the study selection.

**Figure 2 jfmk-06-00012-f002:**
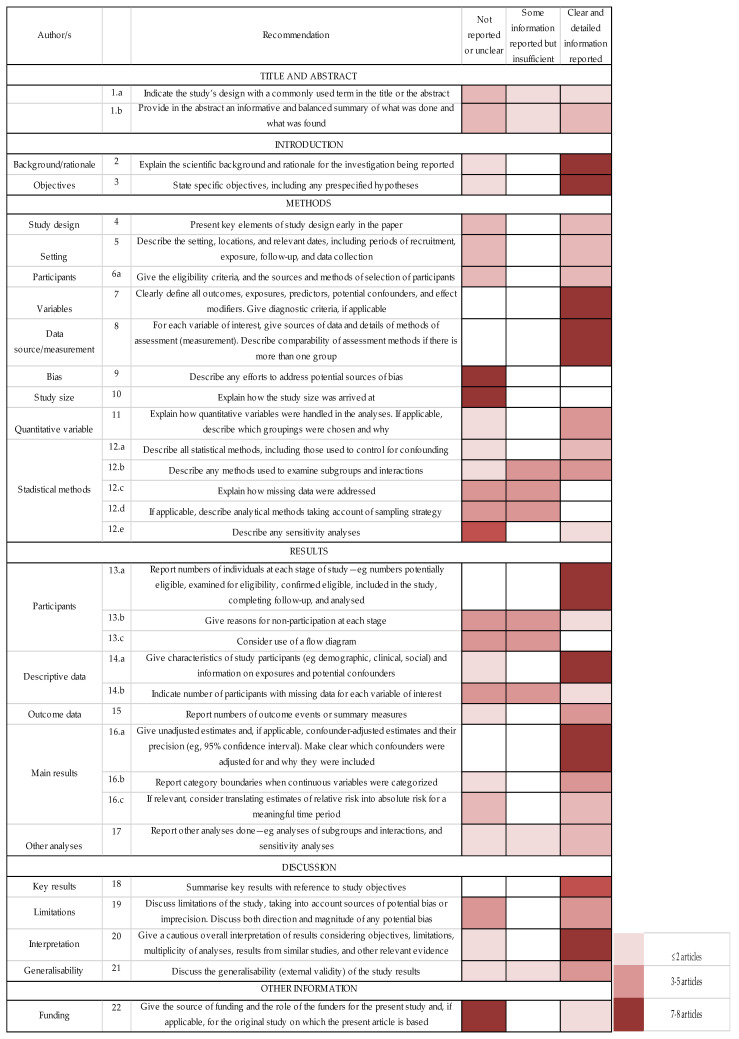
Assessment of reporting completeness and quality of included studies (STROBE).

**Table 1 jfmk-06-00012-t001:** Summary of the studies included in this review.

Authors, Year	Study	Sample Size	Mean Age (Range or SD)	Sport	Grade Impact * Based on Criteria [30]	Main Outcome (UI Tools)	% UI	Secondary Outcomes	Secondary Outcomes Results
Eliasson et al., 2002 [31].	Cross- sectional	*n* = 35	15 (12–22)	Trampoline	3	Pad-weighting test	80 mean leakage of 28 g	Muscular strength with perineometer	23 of 27 diodes on the perineometer for 6 s and 20 for 30 s, 30 cm H2O of intravaginal squeeze pressure
Carls et al., 2007 [18]	Cross- sectional	*n* = 86	17 (14–21)	High impact sports	3	The Bristol Female Lower Urinary Tract Symptoms Questionnaire	28	Educational prevention and treatment of UI	90% had never heard of pelvic muscle exercises (Kegels)
Parmigiano et al., 2014 [19]	Cross- sectional	*n* = 148	15 (2.0)	Soccer, handball, basketball, wrestling, judo, track and field, swimming, boxing	2,3	Pre-participation gynecological examination (PPGE)	Total = 18.2 Track and Field = 14.30; Basketball = 8.30; Boxing = 25; Soccer= 11.60; Handball = 6.40; Judo = 33.30; Swimming = 16.70	Eating attitudes test	15% risk of eating disorders; 89.9% were not familiar with the occurrence of UI in athletes; 87.1% would not mention to coach.
Fernandes et al., 2014 [32]	Cross- sectional	*n* = 35	15.6 (12–19)	Soccer	2	Urinary Incontinence short form (ICIQ-UI SF)	62.8	The pad test and King’s Health Questionnaire (KHQ)	35.2 score in the General Health domain; 37.3 in the emotions domain; 26.5 in the Sleep/Energy domain.
Da Roza et al., 2015 [33]	Cross- sectional	*n* = 22	18.1 (3.4)	Trampoline	3	Urinary Incontinence short form (ICIQ-UI SF)	72.7	Amount of urinary loss, frequency of involuntary loss	93.7% self-classified as moderate amount of UI; frequency of UI once a week or less.
Almeida et al., 2015 [34]	Cross- sectional	*n* = 67	18 (5)	Volleyball, judo, gymnastics, trampoline, swimming	2,3	Urinary Incontinence short form (ICIQ-UI SF)	Total = 52.2 Volleyball = 43.5; Trampoline = 88.9; Swimming = 50; Judo = 44.4	Fecal Incontinence Severity Index, Female Sexual Function Index, vaginal symptoms and pelvic organ prolapse symptoms (ICIQ-VS)	Involuntary loss of gas: 64.6% athletes, 58.5% nonathletes; POP: 0% athletes, 2% nonathletes; dyspareunia: 13.8% athletes, 21.9% nonathletes; 31.4% athlete’s strategy: “Emptying the bladder before training”; 52.0% nonathlete’s strategy: “Emptying the bladder before leaving the house”.
Logan et al., 2017 [14]	Pilot study: Cross-sectional	*n* = 44	(13–17)	Cross-country, track and field field-hockey, soccer	2,3	Urinary Incontinence short form (ICIQ-UI-SF)	48	Identify risk factor	32% vigorous exercise, 34% during laughter, 14% activities of daily living (ADLs).
Dobrowolski et al., 2019 [12]	Cross- sectional	*n* = 89	16 (15–21) *	Rope skipping	3	Prevalence of SUI 11-point Likert scale (0–10)	75	Quality of life (ICIQ-SF), non-validated sport-specific questionnaire inspired by (IIQ-7)	21% indicated an overall interference of SUI with RS as moderate or greater; a slight impact of SUI on their overall quality of life. Female athletes managed SUI with containment products, fluid limitation, and timed voiding.
Gram and Bø 2020 [20]	Cross- sectional	*n* = 107	14.5 (1.6)	Rhythmic gymnastics	3	Urinary Incontinence short form (ICIQ-UI SF)	31.8	Triad-specific self-report questionnaire Beighton score	46.7% hypermobile; 9.3% disordered eating; 29.4% afraid of visible leakage; 14.7% afraid leakage would happen again; 69.1% had never heard about the pelvic floor.

* Mean (IQR); (IQR Interquartile range).

**Table 2 jfmk-06-00012-t002:** Summary of participants’ characteristics.

	Weight (kg)	BMI (kg/m^2^)	Hours of Training/wk
Eliasson et al. [31]	50 (42–60) *	20.3 (19–23) *	-
Carls et al. [18]	-	-	-
Parmigiano et al. [19]	-	21.6 (2.8)	10.9 (4.0)
Fernandes et al. [32]	-	NP	-
Da Roza et al. [33]	55.0 (4.9)	20.4 (1.3)	11.3 (2.7)
Almeida et al. [34]	-	21.7 (2.6) **	19.0 (6.3)
Logan et al. [14]	-	-	-
Dobrowolski et al. [12]	-	21 (20–23) **	6 (4–6)
Gram & Bø [20]	-	18.9 (2.2)	15.7 (7.8)

* Mean (IQR); ** Median (IQR Interquartile range).

## Data Availability

Not applicable.

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
