# Peer review of "The Prevalence of Urinary Incontinence among Adolescent Female Athletes: A Systematic Review"

_jfmk, 2021, doi:10.3390/jfmk6010012_

Round 1

Reviewer 1 Report

Thank you for the opportunity to read and ameliorate this nice and interesting manuscript reporting data on stress urinary incontinence on younger female athletes.

The paper is well done as well as the research. The English is ok but it is possible to find some errors here and there in the text. I gave some corrections and a important change on the kind of SUI are authors talking about. Finally the tables need to be improved.

Here are my comments:

In the introduction it should mentioned Pure Stress Urinary Incontinence (PSUI) due to the fact that young athletes may have only this kind of stress urinary incontinence. The other incidence data may give a general evaluation of SUI incidence but can not be used for younger patients. Please read and cyte "Pure stress urinary incontinence: analysis of prevalence, estimation of costs, and financial impact." BMC Urol. Rubilotta et al. 2019 Jun doi: 10.1186/s12894-019-0468-2. 

page 2, line 53: you have written "Not only can UI impact sports performance [19], but also this condition may also affect their quality of life and ultimately can lead to sport drop out"... please rephrase as follow: UI can affect quality of life, as well as impact sports performance leading to sport drop out.

Table 1 is confusing, the column of Year must be done better (I suggest to use a single column for authors and year), but also in others columns it is not well readable, and last but not least don't use the center but put the text in left side. Please re-edit it.

Table 2 must be re-edit. please fill authors on left side and not centered.

in the text authors use verbs both in the past tense (which is right to use) and in the present tense which is wrong. It is recommended to correct this inconsistency of tenses in verbs. examples are in line 138 "explain", line 141 "describe "...

line 214 "Parmigiano reported that 15% of their sample was at risk for an eating disorders and others noted 9.3% of adolescent rhythmic gymnasts were at risk for disordered eating[19]." please rephrase...

Limitations are at line 221 but should be putted at the end of discussion. Moreover authors should discuss also in the low sample sizes reported in the selected articles (that may be another limitation). Finally, due to the large use of Questionnaires to evaluate SUI it should be also debated the potential limits of these tools in Youngers. Indeed some authors used girls of 12-13 years old.. are we sure these children did understood clearly what asked? were the used questionnaire validated for children? you should debate this..

Author Response

Dear Editor, the authors appreciate the reviewer’s comments. Indeed, they helped us improve the manuscript. We addressed all concerns and included a point-by-point response:

Reviewer 1:

Thank you for the opportunity to read and ameliorate this nice and interesting manuscript reporting data on stress urinary incontinence on younger female athletes.

The paper is well done as well as the research. The English is ok but it is possible to find some errors here and there in the text. I gave some corrections and a important change on the kind of SUI are authors talking about. Finally the tables need to be improved.

Here are my comments:

In the introduction it should mentioned Pure Stress Urinary Incontinence (PSUI) due to the fact that young athletes may have only this kind of stress urinary incontinence. The other incidence data may give a general evaluation of SUI incidence but can not be used for younger patients. Please read and cyte "Pure stress urinary incontinence: analysis of prevalence, estimation of costs, and financial impact." BMC Urol. Rubilotta et al. 2019 Jun doi: 10.1186/s12894-019-0468-2. 

Thank you. We have mentioned this in the introduction and added the following reference to the text: BMC Urol. Rubilotta et al. 2019 Jun doi: 10.1186/s12894-019-0468-2. 

page 2, line 53: you have written "Not only can UI impact sports performance [19], but also this condition may also affect their quality of life and ultimately can lead to sport drop out"... please rephrase as follow: UI can affect quality of life, as well as impact sports performance leading to sport drop out.

Thank you. We’ve rephrased line 53.

Table 1 is confusing, the column of Year must be done better (I suggest to use a single column for authors and year), but also in others columns it is not well readable, and last but not least don't use the center but put the text in left side. Please re-edit it.

Thank you. We revised the table.

Table 2 must be re-edit. please fill authors on left side and not centered.

Thank you. We revised the table.

in the text authors use verbs both in the past tense (which is right to use) and in the present tense which is wrong. It is recommended to correct this inconsistency of tenses in verbs. examples are in line 138 "explain", line 141 "describe "...

We have reviewed the grammar in the text and corrected the tense of the verbs.

line 214 "Parmigiano reported that 15% of their sample was at risk for an eating disorders and others noted 9.3% of adolescent rhythmic gymnasts were at risk for disordered eating[19]." please rephrase...

Thank you. We edited line 214.

Limitations are at line 221 but should be putted at the end of discussion. Moreover authors should discuss also in the low sample sizes reported in the selected articles (that may be another limitation). Finally, due to the large use of Questionnaires to evaluate SUI it should be also debated the potential limits of these tools in Youngers. Indeed some authors used girls of 12-13 years old.. are we sure these children did understood clearly what asked? were the used questionnaire validated for children? you should debate this..

Thank you. We moved the limitations to the end of the discussion and added the suggested limitations.

Reviewer 2 Report

This is an excellent review on the Prevalence of Urinary Incontinence among Adolescent Female Athletes. I found the paper to be well written with an ample bibliography backed up by quality analysis and extensive discussion.

I have only one comment. The paper speaks about urinary incontinence without making a distinction between stress and urge urinary incontinence. Since the review was conducted on adolescent females, it is natural to assume that they refer exclusively to stress urinary incontinence. Nevertheless, it would be good to have confirmation because it would be useful to know if the type of physical activity the subjects underwent could result also in urge urinary incontinence, or indeed other types.

Author Response

Dear Editor, the authors appreciate the reviewer’s comments. Indeed, they helped us improve the manuscript. We addressed all concerns and included a point-by-point response:

Reviewer 2:

I have only one comment. The paper speaks about urinary incontinence without making a distinction between stress and urge urinary incontinence. Since the review was conducted on adolescent females, it is natural to assume that they refer exclusively to stress urinary incontinence. Nevertheless, it would be good to have confirmation because it would be useful to know if the type of physical activity the subjects underwent could result also in urge urinary incontinence, or indeed other types.

Thank you. In line with reviewer 1, we have addressed this concern in the introduction noting that young female athletes display pure stress incontinence symptoms.
